# ALOHA Unleashed: A Simple Recipe for Robot Dexterity

**Tony Z. Zhao,**[*] **Jonathan Tompson, Danny Driess, Pete Florence,**
**Kamyar Ghasemipour, Chelsea Finn, Ayzaan Wahid**[*]

Google DeepMind

**Abstract:** Recent work has shown promising results for learning end-to-end robot policies using imitation learning. In this work we address the question of how far can we push imitation learning for challenging dexterous manipulation tasks. We show that a simple recipe of large scale data collection on the ALOHA 2 platform, combined with expressive models such as Diffusion Policies, can be effective in learning challenging bimanual manipulation tasks involving deformable objects and complex contact rich dynamics. We demonstrate our recipe on 5 challenging real-world and 3 simulated tasks and demonstrate improved performance over state-of-the-art baselines. The project website and videos can be found at aloha-unleashed.github.io.

**Keywords:** Imitation Learning, Manipulation

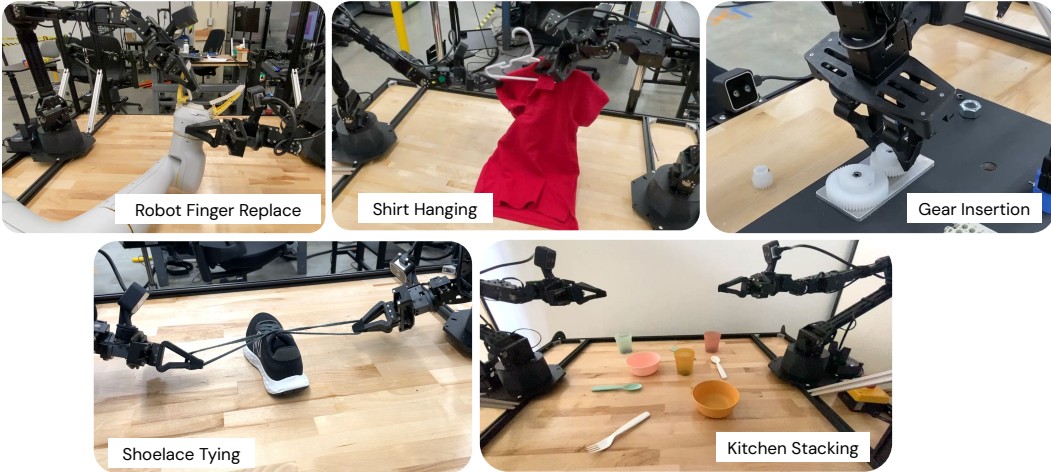

Figure 1: ***ALOHA Unleashed*** trains a transformer encoder-decoder architecture with a diffusion loss to learn highly dexterous bimanual manipulation tasks like hanging a shirt and tying shoelaces.

## 1 Introduction

Dexterous manipulation tasks such as tying shoe laces or hanging t-shirts on a coat hanger have traditionally been seen as very difficult to achieve with robots. From a modeling perspective, these tasks are challenging since they involve (deformable) objects with complex contact dynamics, require many manipulation steps to solve the task, and involve the coordination of high-dimensional robotic manipulators, especially in bimanual setups, and generally often demand high precision. In recent years, imitation learning has been established as a recipe for obtaining policies that can solve a wide variety of tasks. However, most of these success stories predominantly concern non-dexterous tasks such as pick and place [1], or pushing [2]. Therefore, it is unclear if simply scaling

---

[*]Equal contribution

8th Conference on Robot Learning (CoRL 2024), Munich, Germany.

up imitation learning is sufficient for dexterous manipulation, since collecting a dataset that covers the state variation of the system with the required precision for such tasks seems prohibitive.

In this paper, we demonstrate that by choosing the appropriate learning architecture combined with a suitable data collection strategy, it is possible to push the frontier of dexterous manipulation with imitation learning. We show on the ALOHA 2 [3] platform that we can obtain policies that are capable of solving highly dexterous, long-horizon, bimanual manipulation tasks that involve deformable objects and require high precision. To achieve this, we develop a protocol to collect data on a scale previously unmatched by any bimanual manipulation platform, with over 26,000 demonstrations for 5 tasks on a real robot, and over 2,000 demonstrations on 3 simulated tasks.

However, we find that the data alone is insufficient. The other key ingredient in our approach is a transformer-based learning architecture trained with a diffusion loss [4, 5]. Conditioned on multiple views, this architecture denoises a trajectory of actions, which is executed open-loop in a receding horizon setting. Results show that non-diffusion based architectures are incapable of solving some of our tasks, despite being previously tuned for the ALOHA platform.

Our experimental evaluation involves 5 real world tasks such as tying shoe laces and hanging clothes on a hanger as well as 3 simulated tasks. We investigate the data complexity and out-of-distribution robustness of our policies. To the best of our knowledge, we are the first to demonstrate an end-to-end policy that can tie shoelaces or hang t-shirts autonomously.

## 2 Related Work

**Imitation learning.** Imitation learning enables robots to learn from expert demonstrations [6]. Early works tackle this problem through the lens of motor primitives [7, 8, 9, 10].

With the development of deep learning and generative modeling, different architectures and training objectives are proposed to model the demonstrations end-to-end. This includes the use of ConvNets or ViT for image processing [11, 12, 13], RNN or transformers for fusing history observations [14, 15, 16], tokenization of the action space [1], generative modeling techniques such as energy-based models [17], diffusion [18] and VAEs [19, 20]. In this work, we push for simplicity of the algorithm, building upon existing imitation learning algorithms. Specifically, we train a transformer-based policy with diffusion loss, inspired by Diffusion Policy [18] and ACT [20]. While unlike previous works, we train on large amounts of data from non-researcher data collectors, who use ALOHA 2 to perform tasks that are both precise and multi-modal.

**Bimanual manipulation.** Bimanual manipulation has a long history in robotics. Early works tackle bimanual manipulation from an optimization perspective, with known environment dynamics [21, 22]. However, obtaining such environment dynamics models can be time-consuming, especially those that captures rich contact or deformable objects. More recently, learning has been incorporated into bimanual systems, including reinforcement learning [23, 24], imitating learning [25, 26, 27, 28, 29], or learning of key points that modulate low-level motor primitives [30, 31, 32]. Previous works have also studied highly dexterous bimanual manipulation tasks, such as knot untying, cloth flattening, or even threading a needle [31, 33, 34]. However, the robots used are much less accessible such as surgical robots from Intuitive. In this work, we use a fleet of low-cost ALOHA 2 systems [3] to study how scaling up data collection itself can already bring significant advancements in robot dexterity, without ultra-precise kinematics and elaborate sensing.

**Scale up robot learning in the real world**. Many works have attempted to scale up robot learning using real-world data collection. Teleoperation is a one way to collect high-quality data, with a human in the loop controlling the robot. Previous works have collected large datasets on single-arm robots using VR controller or haptic device [13, 35, 16, 36, 37], demonstrating generalization to novel scenes and objects. Alternatively, the robot can also be programmed [38] or controlled by Reinforcement Learning (RL) algorithms [39] to collect data autonomously, reducing the need of human supervision. Another way to collect expert data is to use wearable or hand-held devices, such as grippers [40, 41, 42], exoskeleton [43], or tracking gloves [44]. This allows scaling up of data collection without requiring full robots. There are also ongoing efforts to combine all the aforementioned datasets, to train a single model that can control multiple robots [45]. In this work, we focus on scaling up the dexterity aspect of robot learning, together with robust handling of

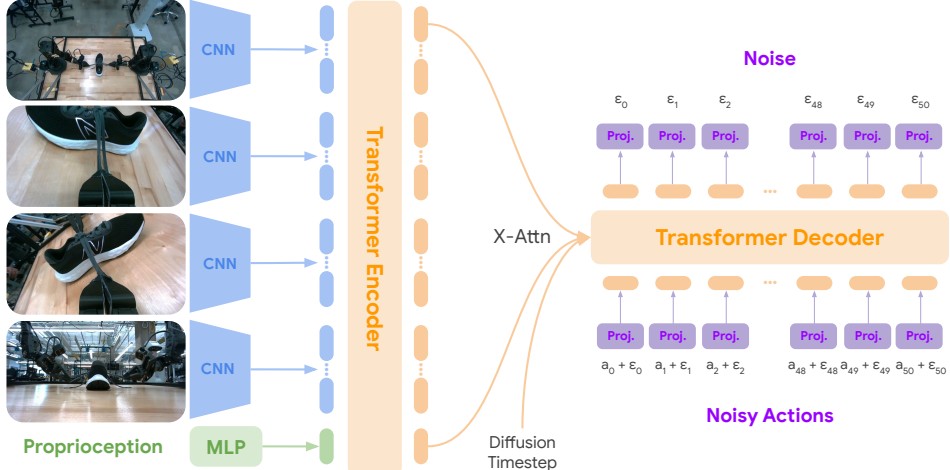

Figure 2: *Policy Architecture.* Each camera view is embedded with a ResNet50 [46]. The Transformer Encoder embeds the observations and produces latent embeddings. The Transformer Decoder takes in noisy actions and cross-attends to the latent embeddings produced by the encoder, outputting the predicted noise. The decoder portion runs 50 times during inference to iteratively denoise an action chunk.

deformable and articulated objects. To the best of our knowledge, we trained the first end-to-end policy that can autonomously tie shoelaces and hang t-shirts.

## 3 Method

We introduce *ALOHA Unleashed*, a general imitation learning system for training dexterous policies on robots. We demonstrate results on ALOHA 2, which consists of a bimanual parallel-jaw gripper workcell with two 6-DoF arms. *ALOHA Unleashed* consists of a framework for scalable teleoperation that allows users to collect data to teach robots, combined with a Transformer-based neural network trained with Diffusion Policy inspired by [18] and [20], which provides an expressive policy formulation for imitation learning. With this simple recipe, we demonstrate autonomous policies on 5 challenging real world tasks: hanging a shirt, tying shoe laces, replacing a robot finger, inserting gears, and stacking randomly initialized kitchen items. We also show results on 3 simulated bimanual tasks: single peg insertion, double peg insertion, and placing a mug on a plate.

### 3.1 Policy

**Diffusion Policy.** The dataset we use has inherent diversity, given that data is collected from multiple operators, contains a variety of teleoperation strategies, and is collected over a long period of time on multiple robotic workcells. This requires an expressive policy formulation to fit the data. We train a separate Diffusion Policy for each task. Diffusion Policy provides stable training and expresses multimodal action distributions with multimodal inputs (4 images from different viewpoints and proprioceptive state) and 14-degree-of-freedom action space. We use the Denoising diffusion implicit models (DDIM) [47] formulation, which allows flexibility at test time to use a variable number of inference steps. We perform action chunking [20] to allow the policy to predict chunks of 50 actions, representing a trajectory spanning 1 second. The policy outputs 12 absolute joint positions, 6 for each 6-dof ViperX arm, and a continuous value for gripper position for each of the two grippers. Since we use action chunks of length 50, the policy outputs a tensor of shape (50, 14). We use 50 diffusion steps during training, with a squared cosine noise schedule from [48].

**Transformer-based architecture.** For our base model, we scale up an architecture similar to the Transformer Encoder-Decoder architecture used in [20]. We use a ResNet50 [46] based vision backbone, with a Transformer Encoder-Decoder [49] architecture as the neural network policy. Each of the 4 RGB images is resized to 480x640x3 and fed into a separate ResNet50 backbone. Each ResNet50 is initialized from an ImageNet [50] pretrained classification model. We take the stage 4 output of the ResNets, which gives a 15 x 20 x 512 feature map for each image. The feature map is flattened, resulting in resulting in 1200 512-dimensional embeddings. We append another

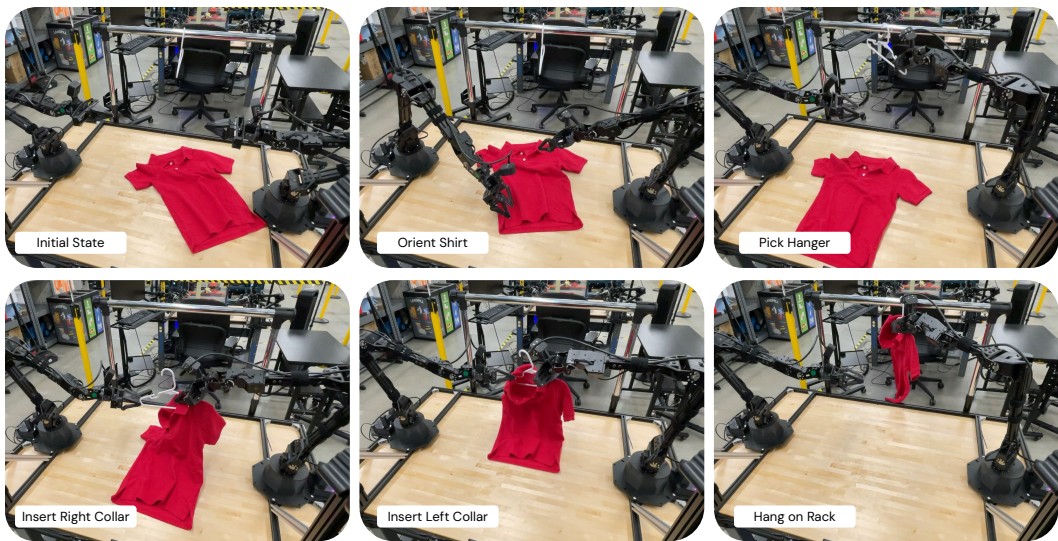

Figure 3: **Shirt Episode.** This task is long horizon, involves a deformable object, and requires several dexterous bimanual behaviors to achieve the end goal of hanging the shirt on the rack.

embedding, which is a projection of the proprioceptive state of the robot, which consists of the joint positions and gripper values for each of the arms, for a total of 1201 latent feature dimensions. We add positional embeddings to the embedding and feed it into a 85M parameter Transformer encoder to encode the embeddings, with bidirectional attention, producing latent embeddings of the observations. The latents are passed into the diffusion denoiser, which is a 55M parameter transformer with bidirectional attention. The input of the decoder transformer is a 50 x 14 tensor, corresponding to a noised action chunk with a learned positional embedding. These embeddings cross-attend to the latent embeddings from the observation encoder, as well as the diffusion timestep, which is represented as a one-hot vector. The transformer decoder has an output dimension of 50 x 512, which is projected with a linear layer into 50 x 14, corresponding to the predicted noise for the next 50 actions in the chunk. In total, the *Base* model consists of 217M learnable parameters. For ablation experiments, we also train a *Small* variant of our model, which uses a 17M parameter Transformer encoder and 37M parameter Transformer decoder, with a total network size of 150M parameters.

**Training details.** We train our models with JAX [51] using 64 TPUv5e chips with a data parallel mesh. We use a batch size of 256 and train for 2M steps (about 265 hours of training). We use the Adam [52] optimizer with weight decay of 0.001 and a linear learning rate warmup for 5000 steps followed by a constant rate of 1e-4.

**Test time inference.** At test time, we first sample a noised action chunk from a gaussian distribution. We gather the latest observations from the 4 RGB cameras and the proprioceptive state of the robot, and pass these through the observation encoder. We then run the diffusion denoising loop 50 times, outputting a denoised action chunk. We find that we do not need the temporal ensembling from [20], and simply execute the 50 actions in the chunk open loop. A full forward pass through the network and iterative denoising process takes 0.043 seconds on a RTX 4090 GPU. Since we run the action chunk open loop, we are able to surpass our target frequency of 50Hz.

## 3.2 Data Collection

ALOHA allows bimanual teleoperation via a puppeteering interface, which allows a human teleoperator to backdrive two smaller leader arms, whose joints are synchronized with two larger follower arms. We collect data on the following 5 tasks:

- **Shirt hanging (Shirt)**: This task requires hanging a shirt on a hanger. The detailed steps include flattening the shirt, picking a hanger off a rack, performing a handover, picking up the shirt, precisely inserting both sides of the hanger into the shirt collar, then hanging the shirt back on the rack. This is a challenging task that requires multiple steps with de-

| Task | Success Rate | Number of Demonstrations |
|------|--------------|--------------------------|
| ShirtEasy | 75% | 8658 (5345 Easy; 3313 Messy) |
| ShirtMessy | 70% | |
| LaceEasy | 70% | 5133 (2212 Easy; 2921 Messy) |
| LaceMessy | 40% | |
| FingerReplace | 75% | 5247 |
| GearInsert-1 | 95% | |
| GearInsert-2 | 75% | 4005 |
| GearInsert-3 | 40% | |
| RandomKitchen-Bowl | 95% | |
| RandomKitchen-Bowl+Cup | 65% | 3198 (216 In-Domain) |
| RandomKitchen-Bowl+Cup+Fork | 25% | |

Table 1: Success rates and number of demonstrations for 5 real tasks, with separate models for each task. Shirt and Lace tasks each have a single model trained on 8658 and 5133 demonstrations, respectively, and are separately evaluated on Easy and Messy variants of the tasks. The GearInsert and RandomKitchen model evaluations are broken down by total progress on the task.

formable manipulation, insertions, and dexterous pick and placing behaviors like hooking and unhooking the hanger from the rack. We construct two variants of this task: **ShirtEasy** has a more constrained initialization, with the shirt flattened and centered on the table. **ShirtMessy** allows the initialization of the shirt to be rotated and crumpled and has significantly more variance in starting location.

- **Shoelace tying (Lace)**: This task requires centering a shoe on the table, straightening the laces, then performing a maneuver to tie the laces in a bow. We construct two variants of this task: **LaceEasy** has a constrained initialization with the shoe centered on the table and laces extended outward. **LaceMessy** allows $\pm45$ degree variance in the angle of the shoe and does not require the laces to be flattened.
- **Robot finger replacement (FingerReplace)**: This task requires removing a robot finger from a slotted mechanism, picking the replacement finger, reorienting the finger, then performing a precise insertion back into the slot with millimeter tolerance.
- **Gear insertion (GearInsert)**: This task requires inserting 3 plastic gears onto a socket with millimeter precision with a friction fit, while ensuring that the gear is fully seated and the teeth on the gear mesh with neighboring gears.
- **Random kitchen stack (RandomKitchen)**: This task requires cleaning up a randomly initialized table by stacking bowls, cups, and utensils and placing the stack at the center of the table.

To scale data collection on these tasks, we create a protocol that allows non-expert users to provide high quality teleoperated demonstrations. Protocol documents (See Appendix B.1) outline instructions for both how to use the robots, and specific instructions for the task being performed. This allows continuous data collection by a pool of 35 operators without oversight by researchers. Using this protocol, we collect over 26k episodes for 5 real tasks, on 10 different robots in 2 different buildings over the course of 8 months. Data collection over a long period of time on multiple robotic workcells presents many challenges. Robots may have differences in hardware assembly such as mounting positions for the robots or cameras, due to either assembly mistakes or general variance. In addition, hardware changes or general wear and tear on robots may change robot dynamics and behavior. Changes across buildings and differences in robot placement contribute to diversity of backgrounds and lighting in RGB images. Collecting data from 35 different operators results in a large amount of variance in behaviors, even with detailed protocol documentation for each task.

## 4 Results

### 4.1 Task Performance

For our main results of our core models, we perform 20 trials on each task using models separately trained on 5 datasets (Shirt, Lace, FingerReplace, GearInsert, and RandomKitchen). An episode

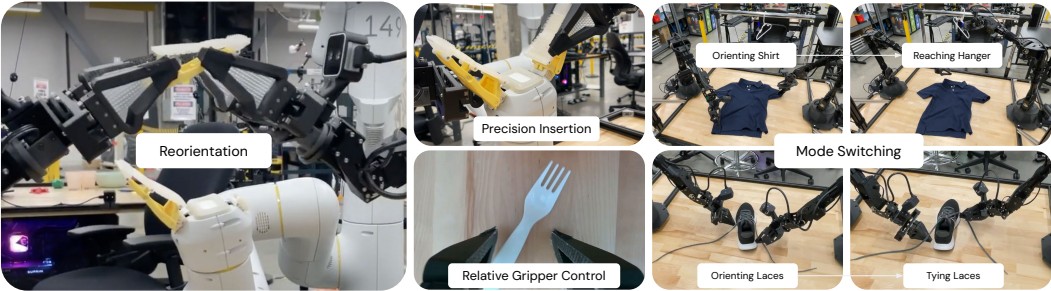

Figure 4: *Learned Behaviors.* We observe interesting behaviors the model is able to learn from the data, including reorientation and precision insertion in FingerReplace, relative gripper control in RandomKitchen, and mode switching behaviors in ShirtMessy and LaceMessy.

terminates either on success or a timeout (120 seconds for ShirtMessy and 80 seconds for other tasks). For GearInsert, we report a more detailed breakdown of task progress, where GearInsert-1 represents successful insertion of at least 1 gear, GearInsert-2 is successful insertion of 2 gears, and GearInsert-3 is successfully inserting all 3 gears. For KitchenStack, we also report a detailed breakdown based on task progress. For GearInsert and KitchenStack, we see that performance decreases with each additional stage, usually due to more fine grained behaviors needed to insert smaller gears or pick thin objects like forks. For all other tasks, we mark a success only if the policy performs all required steps, with no partial success.

## 4.2 Learned Dexterous Behaviors

In this section we highlight dexterous behaviors that the policy is able to learn from the data.

While collecting data for these tasks, operators perform many **bimanual behavior primitives**, like handovers for reorientation, and *view augmentation* with wrist cameras. For example, FingerReplace requires reorienting the finger after picking it off the table to align the finger's direction for insertion. We see that the policy learns many coherent reorientation behaviors from multiple starting positions of the robot finger. Though we find that reorienting is fairly robust, we see failures for starting positions that aren't well represented in the dataset, such as the finger being flipped upside down, suggesting that it may be necessary to explicitly collect more diverse examples of reorientation. In FingerReplace, the policy also learns a bimanual *view augmentation* strategy that is present in the data, where operators use the wrist camera from the unsused arm to provide augmented RGB input for the policy to better perform the precise insertion, which is less visible from other views.

We see many instances of **recovery behaviors and retries** in all tasks. For example, on the shirt tasks we see instances of the shirt falling off the hanger, and the policy recovering and replacing the shirt on the hanger. We also see instances of retry behavior during insertions such as in GearInsert and FingerReplace, where the policy reorients and recovers from failed inserts.

The policies perform **relative gripper control** to accomplish precise picking behavior involved in all tasks. This is especially apparent in RandomKitchen, which requires picking thin objects from the table from a wide variety of initial states. Robots in our ALOHA 2 fleet are uncalibrated and may have differences in robot and camera mounting positions. We speculate that though the policy receives RGB and full proprioceptive state, policies may be learning to perform reactive relative gripper control from visual feedback to generalize across robots.

On several long horizon tasks, we observe **mode switching** behaviors. For example, on ShirtMessy, the policy changes from flattening the shirt on the table to beginning the reach for the hanger. Similarly on LaceMessy, the policy switches from straightening the shoe to the loop-tying phase of the episode.

GearInsert and FingerReplace both require millimeter-accuracy **precision insertions**. GearInsert, in particular, requires a tight friction fit to properly align and insert the gear all the way into the shaft. We find it surprising that despite low-precision robotic arms and lack of force-torque feedback, our policies are able to perform these tasks with only visual feedback.

|                         | ShirtEasy | ShirtMessy | Number of Demonstrations |
|-------------------------|-----------|------------|--------------------------|
| **Shirt-100%**          | 75%       | 70%        | 8,658                    |
| **Shirt-75%**           | 75%       | 70%        | 6,493                    |
| **Shirt-50%**           | 85%       | 20%        | 4,329                    |
| **Shirt-25%**           | 30%       | 0%         | 2,164                    |
| **Shirt-25%-LongFilter**   | 30%    | -          | 1,623                    |
| **Shirt-25%-MediumFilter** | 55%    | -          | 1,082                    |
| **Shirt-25%-ShortFilter**  | 40%    | -          | 541                      |

Table 2: Success rates and number of demonstrations for Shirt data ablations. **Data quantity:** We evaluate model performance on both ShirtEasy and ShirtMessy by randomly sampling 75%, 50%, and 25% of data. **Data filtering:** We evaluate performance on ShirtEasy. We first randomly sample 25% of the dataset, then apply further filtering based on 75th, 50th, and 25th percentiles of episode duration.

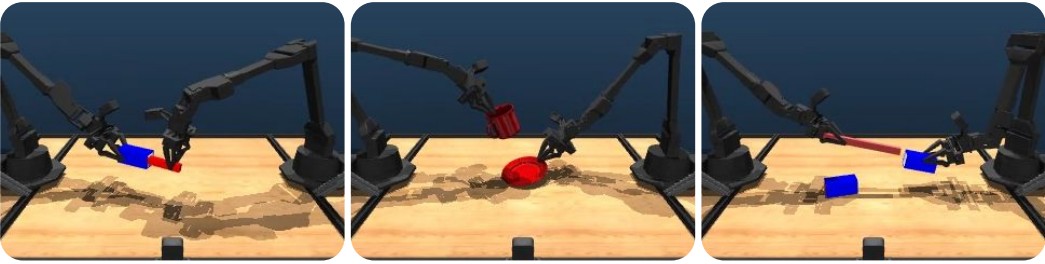

Figure 5: *Simulated Tasks.* From left to right: **SingleInsertion**, which requires inserting the red peg into the blue socket. **MugOnPlate**, with the mug and plate randomly initialized on the table. **DoubleInsertion**, which requires inserting the red peg into two different sockets on either end.

### 4.3 Ablations

We perform several experiments to determine the importance of quantity and quality of demonstration data. All experiments below are run with the *Small* 150M parameter variant of the model.

**Data quantity.** How does task performance vary depending on number of demonstrations? For the Shirt tasks, we train policies on 100%, 75%, 50%, and 25% of data. We find that to an extent, performance of policies trained on less data perform similarly to policies train on all data for the ShirtEasy task. However, policies trained on less data are clearly worse at ShirtMessy. We hypothesize that ShirtMessy requires more demonstrations to learn the dynamic behaviors required for rearranging and flattening the shirt.

**Data filtering.** We observe that shorter episodes collected by operators tend to have less mistakes during the trajectory. We therefore perform data filtering based on the episode duration for ShirtEasy. For this task, we first take a random sample of 25% of the total dataset, resulting in a total of 2164 episodes. In this low data regime, we then train models on the following splits: 1) all episodes, 2) shortest 75% of episodes (shorter than 43s) 3) shortest 50% of episodes (shorter than 29s), 4) shortest 25% of episodes (shorter than 20s). We see on ShirtEasy that performance improves after some amount of data filtering, improving from 30% success when trained on all episodes, to 55% success when trained on the shortest 50% of episodes. However, when using the shortest 25% of episodes (only 541 episodes, though usually mistake-free demonstrations), performance dips to 40%. We speculate that finding a good balance between number of raw demonstrations and demonstrations of varying quality is important. While clean, high quality demonstrations are important for modeling the best behaviors, some amount of suboptimal data may also be necessary, since this data contains recovery and retry behaviors that can help the policy.

**Diffusion vs. L1 Regression Loss.** Since our Transformer-based architecture is very similar to [20], we compare the diffusion loss to a L1 regression loss, which makes the system much closer to ACT. We compare performance on ShirtEasy, ShirtMessy, and the simulated environments. Despite having a well-tuned Action chunking + L1 regression implementation with a 150M parameter model, we observe 25% success on on ShirtMessy compared to 70% for the similar sized Diffusion Policy.

**Simulation experiments.** We compare diffusion and L1 regression loss on 3 simulated bimanual tasks using the MuJoCo Menagerie [53] [54] model from ALOHA 2. We teleoperate the simulated environments as described in [3] to collect human demonstrations for each task, and train Diffusion

| Task | DP (S) | DP (XS-LowRes) | ACT (XS-LowRes) | Num Demos |
|------|--------|----------------|-----------------|-----------|
| SingleInsertion (sim) | 72 | 58 ±3 | 32 | 522 |
| DoubleInsertion (sim) | 60 | 48 ±2 | 58 | 201 |
| MugOnPlate (sim) | 80 | 74 ±0 | 40 | 550 |

| Task | DP (S) | ACT (150M) | Num Demos |
|------|--------|------------|-----------|
| ShirtMessy (real) | 70 | 25 | 8658 |

Table 3: Comparison between Diffusion Policy (DP) and ACT L1 regression loss. For diffusion sim experiments, in addition to the Small (S) model, we run a smaller scale XS-LowRes variant which uses a single vision encoder for each camera and resizes images to 256x256. Comparisons between DP and ACT are done using XS-LowRes. See Appendix A for more details and sim comparisons.

Policies and an ACT L1 regression loss baseline on the datasets. Simulation results are reported over 50 rollouts. For Diffusion Policy (XS-LowRes) models, we run rollouts with 3 seeds. Each episode has a different initialization of object positions. We observe that Diffusion Policy outperforms ACT (for XS-LowRes) for all tasks except DoubleInsertion. See Figure 5 for descriptions of tasks and Appendix A for more analysis.

### 4.4 Generalization

While our core models are only trained per-task, we do observe some promising signs of generalization from our models. In the Shirt tasks, we observe successful rollouts of the model on unseen shirts which are quite different from shirts seen in the training data. Shirts seen in the train set were only kids sizes with short sleeves and red, white, blue, navy, and baby blue colors, while the unseen shirt is a gray adult men's size with long sleeves. We also observe successful rollouts of the Shirt model on an unseen robot in a completely different building (home environment with white wall as background instead of the industrial lab background seen in the train set).

We push the boundaries of our model by measuring our model's generalization ability for the ShirtMessy task, for which we have 3,113 demonstrations in the train set. The state space of the task, however, is still large since the deformable shirts may take many configurations. We observe that the model can handle initializations of the shirt that are ±60 degrees tilted, wrinkled, and right-side-up on the table, with the model learning good behaviors for flattening and centering the shirt given this configuration. We observe that the model usually fails to recover from shirts being 180 degrees or face-down on the table, since there are no instances of this in the training set. Similarly on the Lace tasks, we are able to learn "straightening" behaviors, however fail to recover for states outside of the train distribution (for example, if the shoe tips over, flips around, or the laces get tangled).

On RandomKitchen, we observe some amount of initial state generalization given that the objects can be initialized anywhere within the robot's task space. In addition, we evaluate this model on a robot which has 216 demonstrations, where the other 2,983 demonstrations are collected in another building with a hardware iteration of ALOHA that had different robot mounting positions.

## 5   Conclusion

We present *ALOHA Unleashed* , a simple recipe for learning dexterous robot behaviors. We collect over 26k demonstrations on the ALOHA 2 platform, and train a Transformer-based Diffusion Policy on the data. We demonstrate dexterous behaviors in both real and simulated environments.

*ALOHA Unleashed* shows that a simple recipe could push the boundaries of bimanual, dexterous behaviors in robot learning. However, this approach is limited in several aspects: policies are trained for only one task at a time, whereas other approaches use a single set of model weights that is conditioned on language or goal images to perform multiple tasks. In addition, the policy replans every 1 second, which may not be fast enough for very reactive tasks. *ALOHA Unleashed* also uses many human demonstrations per task, which are time consuming to collect.

We hope to extend *ALOHA Unleashed* to expand the number of tasks using a single model that can perform multiple tasks, add modeling improvements to perform more reactive tasks, and continue to improve data complexity to reduce the amount of data required to learn dexterous behaviors.

**Acknowledgments**

We thank Spencer Goodrich for leading hardware development, mechatronics, and maintenance on our ALOHA 2 fleet. We thank Thinh Nguyen for hardware engineering and maintenance. We thank Travis Armstrong for leading operations. We thank Jaspiar Singh and Justice Carbajal for leading data collection operations. We thank Scott Lehrer, Rochelle Dela Cruz, Tomas Jackson, Jodexty Therlonge, Joel Magpantay, Gavin Gonzalez, Tram Pham, Samuel Wan, Sphurti More, Rosario Jauregui Ruano, April Zitkovich, Alex Luong, Cheri Tran, Emily Perez, Sangeetha Ramesh, Brianna Zitkovich, Utsav Malla, Grecia Salazar Estrada, Elio Prado, Kevin Sayed, Joseph Dabis, Dee M, Eric Tran, Diego Reyes, Jodilyn Peralta, Celeste Barajas, Gabriela Taras, Sarah Nguyen, Lama Yassine, Steven Vega, and Clayton Tan for contributing to data collection.

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

# A    Additional Simulation Experiments

**Sim Experiment Protocol**. For results in Table 3, we run 50 evaluations for checkpoints taken every 100k steps during training. We report the maximum evaluation score over all checkpoints. The Diffusion Policy Small (DP (S)) variant is run for 2M train steps to match real, while XS-LowRes variants are run for 1M train steps.

**Sim Analysis** We see that ACT can outperform Diffusion in certain settings. However for most tasks, and in particular the real Shirt task, we find that Diffusion Policy outperforms ACT. Qualitatively, we find that ACT with a smaller network is easier to tune and often times performs better in a low data regime. This is consistent with the higher score for XS-LowRes variant on DoubleInsertion. Generally when onboarding a new task, we find it helpful to run ACT to get signs of life before moving onto Diffusion Policy after we collect larger, more multi-modal datasets from many operators.

**Evaluation Curves**. We report full evaluation curves of a simulation run. Due to resource constraints, for sim experiments we use an Extra Small Low Resolution (XS-LowRes) variant of the model which uses a single vision encoder for each camera, with images resized to 256x256.

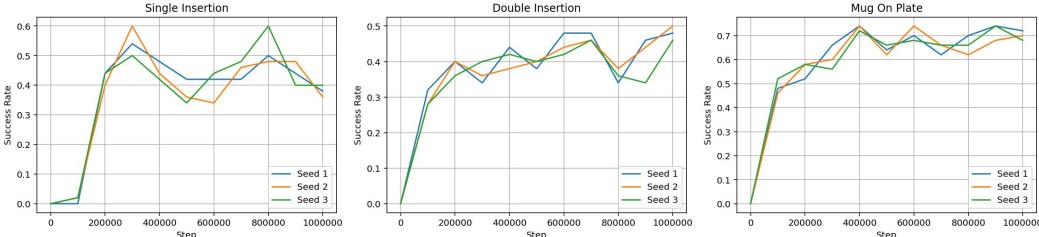

Figure 6: *Full sim evaluation curves*. We run 50 evaluations per checkpoint every 100k steps to produce curves of evaluation performance for the XS-LowRes model over the course of training.

**Chunk Size**. We run a model with chunk size 10 for the simulated single insertion task. The best performance over a single seed is 66, which is higher than the average score of $58 \pm 3$ over 3 seeds of the equivalent model with chunk size 50 (See DP (XS-LowRes) in Table 3). In our experience, tuning the chunk size may lead to better results depending on the task. However this is highly task dependent and also requires tuning other hyperparameters. For our real experiments, we tuned our experiments with a chunk size of 50 as this showed good qualitative performance on all tasks.

**Diffusion Steps**. To measure the effect of certain choices for diffusion sampling, we run a model with 50, 25, and 2 diffusion steps at inference time with the DDIM sampler. Evaluation curves in Figure 7 show that reducing the number of diffusion steps has little effect on this sim task. On real tasks, we run with the full 50 steps since this did not meaningfully affect the inference time in our setup. Given evaluation scores are similar, using fewer diffusion steps at test time may be a desired strategy in more compute constrained environments.

|  | **Single Insertion** | | | **Double Insertion** | | | **Mug On Plate** | | |
|---|---|---|---|---|---|---|---|---|---|
| **Step** | **Seed 1** | **Seed 2** | **Seed 3** | **Seed 1** | **Seed 2** | **Seed 3** | **Seed 1** | **Seed 2** | **Seed 3** |
| 0 | 0.00 | 0.00 | 0.00 | 0.00 | 0.00 | 0.00 | 0.00 | 0.00 | 0.00 |
| 100000 | 0.00 | 0.02 | 0.02 | 0.32 | 0.28 | 0.28 | 0.48 | 0.46 | 0.52 |
| 200000 | 0.44 | 0.40 | 0.44 | 0.40 | 0.40 | 0.36 | 0.52 | 0.58 | 0.58 |
| 300000 | 0.54 | 0.60 | 0.50 | 0.34 | 0.36 | 0.40 | 0.66 | 0.60 | 0.56 |
| 400000 | 0.48 | 0.44 | 0.42 | 0.44 | 0.38 | 0.42 | 0.74 | 0.74 | 0.72 |
| 500000 | 0.42 | 0.36 | 0.34 | 0.38 | 0.40 | 0.40 | 0.64 | 0.62 | 0.66 |
| 600000 | 0.42 | 0.34 | 0.44 | 0.48 | 0.44 | 0.42 | 0.70 | 0.74 | 0.68 |
| 700000 | 0.42 | 0.46 | 0.48 | 0.48 | 0.46 | 0.46 | 0.62 | 0.66 | 0.66 |
| 800000 | 0.50 | 0.48 | 0.60 | 0.34 | 0.38 | 0.36 | 0.70 | 0.62 | 0.66 |
| 900000 | 0.44 | 0.48 | 0.40 | 0.46 | 0.44 | 0.34 | 0.74 | 0.68 | 0.74 |
| 1000000 | 0.38 | 0.36 | 0.40 | 0.48 | 0.50 | 0.46 | 0.72 | 0.70 | 0.68 |

Table 4: Raw evaluation results for all sim seeds from experiments in Figure 6.

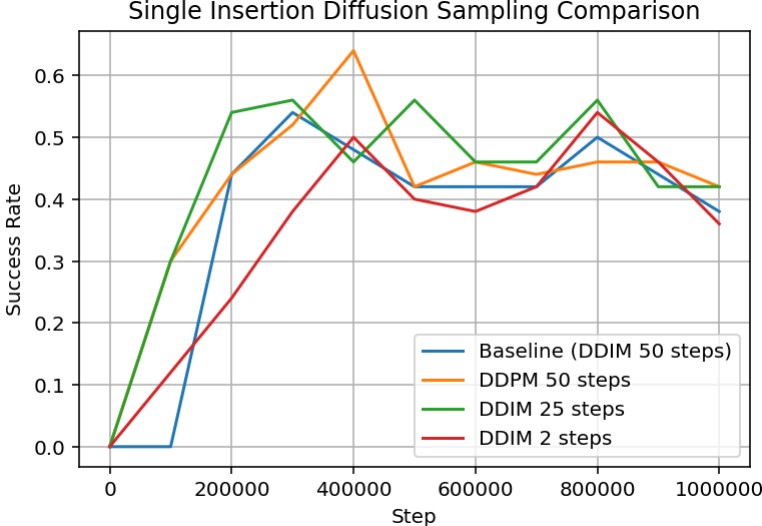

Figure 7: ***Comparison of Diffusion Sampling***. For each checkpoint, we compare different sampling strategies for the SingleInsertion task.

## B    Dataset Details

### B.1    Protocol Documents

We provide the protocol instructions given to operators to collect the 5 tasks. Operators are given these instructions and are instructed to collect a few (5 - 10) test episodes that researchers review for quality. For a few tasks such as ShirtMessy and Lace, we provide a short in person tutorial to an operator to teach and revise the best strategy for performing the task. After the initial test episodes, operators collect the remaining data.

| Task | Protocol Document |
|---|---|
| Shirt | https://aloha-unleashed.github.io/assets/shirt_protocol.pdf |
| FingerReplace | https://aloha-unleashed.github.io/assets/finger_protocol.pdf |
| Lace | https://aloha-unleashed.github.io/assets/lace_protocol.pdf |
| GearInsert | https://aloha-unleashed.github.io/assets/gear_protocol.pdf |
| RandomKitchen | https://aloha-unleashed.github.io/assets/kitchen_protocol.pdf |

### B.2    Initial State Distributions

We show in Figure 8 visualizations of 16 logged initial states for an evaluation of RandomKitchen demonstrate the initial state variance for this task. For this and other tasks, see Appendix B.1 for details on how initial states were varied during data collection. These were the state variations used for evaluations of all tasks.

## C    Hardware Variance

To highlight variations between robots in our fleet, we make measurements of the base positions of 12 ALOHA 2 robots. Note that these measurements were made several months after data collection and evaluations in this paper, though a similar amount of discrepancies between robots existed when experiments were performed.

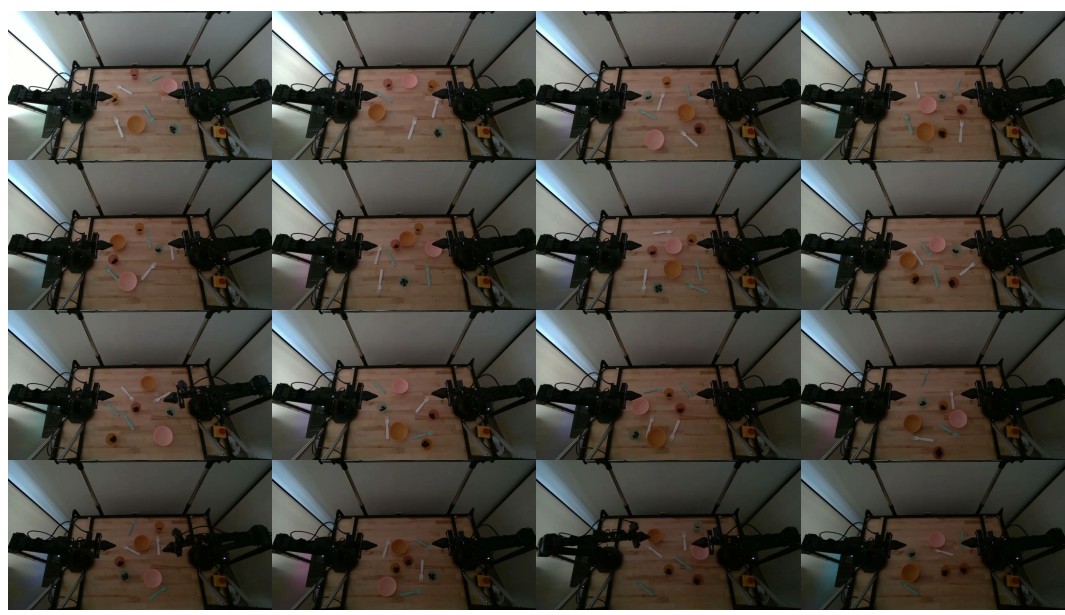

Figure 8: *A set of representative initial states for RandomKitchen evaluations*.

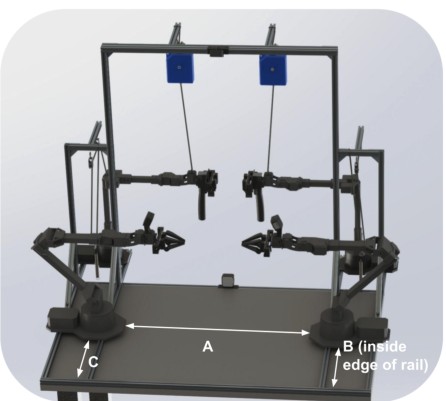

Figure 9: Diagram of an ALOHA 2 workcell, with labels for segments measured for workcell variance. See Table 5 for actual measurements.

|  | **A** | **B** | **C** |
|---|---|---|---|
| **aloha1** | 731.5 | 278 | 277.5 |
| **aloha2** | 7331 | 273 | 274 |
| **aloha3** | 735 | 275 | 276 |
| **aloha4** | 732.5 | 274 | 274 |
| **aloha5** | 732 | 278.5 | 279.5 |
| **aloha6** | 731.5 | 275 | 275 |
| **aloha7** | 736 | 273.5 | 274.5 |
| **aloha10** | 733.5 | 268 | 268.5 |
| **aloha14** | 734 | 274 | 274 |
| **aloha18** | 736 | 274 | 273.5 |
| **aloha19** | 735 | 274 | 274 |
| **aloha22** | 735 | 277 | 278.5 |

Table 5: Measurements of robot base positions. Measurements are in millimeters. See Figure 9 for diagram showing which segments were measured.

