# OpenReview forum: "ALOHA Unleashed: A Simple Recipe for Robot Dexterity"
_robot-learning.org/CoRL/2024/Conference — CoRL 2024_

### Official Review · Reviewer_zByN · 2024-07-20
**Aloha Unleashed**

**Originality:** 3
**Technical Quality:** 4
**Clarity Of Presentation:** 5
**Potential Impact:** 3
**Recommendation:** 4
**Confidence:** 4

**Review:**

This submission read like a technical report. The key scientific insight is that scaling dexterous manipulation for very challenging tasks, like tying shoelaces, works very well when sufficient quantities of training data are collected.

For the purposes of the review, I focused mainly on the questions that I still have outstanding, as such I am just going to copy the first ones into this review panel in order to not create an excessively verbose text that is of marginal interested to all the parties involved.

--- Beginning of Questions

1. According to Fig 2. only the diffusion time steps are provided to the policy, not the episode time step, this would mean that in a temporally symmetric task, e.g. moving to light switch, switching it and moving back, the current setup without state velocities or images from previous time steps the setup would not be expected to work. What do the authors suggest to best resolve this problem?

2. Is there any obvious reason why relative Cartesian control (e.g. in the gripper coordinate frame) would not work? This should be tested.

3. Why is training done for 2M steps fixed. Was there a validation loss, and is there overfitting? Does a validation loss not give a good indication of final task performance? This should easily be testable in simulation with existing snapshots.

**Quality Of The Limitations Section:**

3

**Questions For Rebuttal:**

4. There are 10 robots used in the experiments, does that mean that 10 robots were used collecting data for each of the tasks? Was evaluation done using one of these, or was there a held-out robot? Was there a statistical difference in test or validation loss between the different robots during training.

5. Just saying that the robots were uncalibrated is a bit weak. It should be possible to quantify this with e.g. mean difference in robot base positions between robot setups. Furthermore, in figure 4 it looks like e.g. the camera position has been intentionally modified in a task dependent manner, e.g. closer for laces.

6. (major point) The use of "visual servoing" to describe the behavior is ambiguous at best. My understanding is that visual servoing systems use closed loop control loops to achieve goal states, which the process of servoing takes them towards. Goal states in this method don't explicitly exist during inference time, and calling it servoing just because goal states are learned would not be so helpful, as this is the case for many learned policies. The aspect that that paragraph describes is relative gripper/object control, which could maybe be described instead as implicit calibration, as the policy measures and adjusts for the gripper position.

7. Why is no augmentation used, especially if implicit calibration is happening (and desirable). This should boost performance. If the robots are symmetric, horizontal flip with left-right robot switching should also be possible.

8. Looking at the high loss samples in training or validation, is there any pattern visible?

Minor Points:
L86: multimodality -> diversity

L256: add "for the"

**Robotics Focus:**

4

**Summary Of Paper:**

Scaling dexterous manipulation with more training data and diffusion policies

**Summary Of Recommendation:**

The paper presents a clear verification of the hypothesis that learned diffusion policies scale for manipulation tasks. The technical quality of this paper is good.:

---

### Official Review · Reviewer_zwLY · 2024-07-20
**Paper Review**

**Originality:** 4
**Technical Quality:** 3
**Clarity Of Presentation:** 3
**Potential Impact:** 3
**Recommendation:** 3
**Confidence:** 4

**Review:**

### Strengths

[1] The paper is well-written and easy to follow. The overall approach and experimental results are presented in a clear and concise manner.

[2] The work demonstrates really impressive results on long horizon dexterous manipulation tasks such as shoe lace tying, shirt hanging, robot finger replacement.

### Weaknesses
[1] Technical Contribution: It is hard to pin-point the exact technical contribution of the paper. The paper is presented as a “recipe for learning dexterous robot behaviors”, however, it seems to be a straightforward application of Diffusion Policy with large scale data collection, which by itself is not a novel contribution.

[2] The authors do not provide any details about the data collection protocol. Since data collection is a main contribution of the paper, it is important to know what specific details of data collection protocol help achieve these impressive results.

[3] Experimental Results: The experimental results are presented at a high level and somewhat hand wavy manner. For instance, the Generalization results in Sec 4.4 have no numerical metrics and are presented using verbiage like “we observe some successful episodes”. It is hard to tell from this whether the model actually generalizes to different scenarios such as different shirt types or shoe laces or just gets lucky in some cases. While it can be understandable that it is hard to do multiple ablations in the real world, there is a lack of rigorous testing even in the simulation experiments. For example, metrics presented in Table 3 do not contain any standard deviation or confidence values over multiple seeds. They are also missing ablations for different components of their policy architecture such as number of actions in the chunk or number of diffusion steps.

[4] In the absence of a proper scaling law analysis (eg. similar to [1]), it is unclear how well the algorithm actually scales with data since separate data is collected for each task.




### References

[1] Springenberg, Jost Tobias, Abbas Abdolmaleki, Jingwei Zhang, Oliver Groth, Michael Bloesch, Thomas Lampe, Philemon Brakel et al. "Offline actor-critic reinforcement learning scales to large models." arXiv preprint arXiv:2402.05546 (2024).

**Quality Of The Limitations Section:**

2

**Questions For Rebuttal:**

Q1: Can the authors provide details of the data collection protocols or what the important features of them are?

Q2: In addition to data filtering based on episode length, are there other methods for data filtering that can help with learning performance?

Q3: Are results in Table 3 reported over multiple seed values or just a single seed?

**Robotics Focus:**

4

**Summary Of Paper:**

The paper presents an imitation learning approach for learning challenging dexterous manipulation tasks using the bimanual Aloha framework. The approach combines Diffusion Policies with large scale data collection and demonstrates scaling to contact rich real-world tasks such as shirt hanging, gear insertion and shoelace tying.

**Summary Of Recommendation:**

My recommendation is based on the fact that even though the presented demos are really impressive, the experimental analysis is missing key details that make it a weaker conference paper. The current draft reads more like a technical report or blog post and additional details are required to be considered for publication.

---

### Official Review · Reviewer_qSxg · 2024-07-21
**Impressively challenging tasks, but concerning trend in amount of human effort required**

**Originality:** 4
**Technical Quality:** 5
**Clarity Of Presentation:** 5
**Potential Impact:** 3
**Recommendation:** 3
**Confidence:** 4

**Review:**

## Strengths:
1. The paper addresses an important question: is scaling up existing BC methods (such as Diffusion Policy) with more demonstrations enough to solve dexterous tasks?
2. The difficulty of the tasks addressed (such as tying shoelaces and hanging a shirt) is impressive and goes beyond what prior work in end-to-end learning has achieved. The videos alone will really inspire the community.
3. If I understand correctly, the full action prediction process, including 50 steps of diffusion denoising, takes 43 ms (Line 128). This is really quite fast for a diffusion-based inference algorithm and transformer-based architecture, so clearly it has been designed well.
4. It is good to see the method’s focus on simplicity, for example leaving out the temporal ensembling from [20].
5. There are interesting insights from ablation studies. For example, using data filtering to balance data quality vs number of demonstrations.

## Weaknesses:
1. One weakness of this work is that the novel technical contribution in terms of developing the method is not very clear. The paper should go into much more detail in how the proposed architecture and method differ from the original Diffusion Policy paper (which included its own transformer architecture), and the many follow-up works on diffusion policy (such as Scaling Up and Distilling Down). Are there new changes to diffusion policy, introduced by this paper, which are key in making it possible to solve these new challenging tasks? If so, a direct ablation on those changes would be helpful. If this is not the case, then I think the paper still works, but it should be clearer that this is more about taking an existing method (diffusion policy with a transformer) and investigating whether adding more demonstrations is enough to solve dexterous tasks.
2. There are some results in the paper which are valuable for the community to know, but also are concerning for the future of this line of work (pixel-to-actions imitation learning). The amount of human effort required for this method to work is very high: 35 human operators collecting 26,000 demonstrations over 8 months, including 8658 demonstrations for the shirt hanging task alone. Additionally, the scaling curve relating performance to the number of demonstrations is also slightly concerning: there is the same 70% success rate on the ShirtMessy task when increasing the number of demonstrations from 6493 to 8658. This suggests that collecting more demonstrations alone may not be enough to achieve product-ready success rates on these very difficult tasks, and that collecting enough demonstrations for a generalist robot may be too expensive. The authors do mention in the future work section that they would like to improve data complexity, but there is not much detail on how this can be done. Stronger motivation is needed in this paper to show that this line of work will continue to have impact in the future. For example, a small-scale comparison against a more data-efficient imitation learning method, such as EquivAct (ICRA 2024), could help motivate the need for the pixel-to-action method proposed in ALOHA Unleashed. ALOHA may perform better on e.g. tasks requiring more precision, even if it requires more demonstrations.
3. This method predicts a sequence of actions (about 1 second in advance) and then executes that sequence open-loop, if I understand correctly, while predicting the next action chunk. This probably leads to smoother motion, but makes the policy much less reactive (up to 1 second of delay). This might limit the method for tasks that require more reactive manipulation, e.g. when the object might slip.

## Minor issues:
1. Line 104: this currently says “resulting in a 1200 dimensional embedding for the 4 images”. It may be a little clearer to say something like “resulting in 1200 512-dimensional embeddings”, or some way to convey that 1200 is the number of vectors, rather than the dimensionality of the vector (which I assume is 512, based on Line 103).

## Update post-rebuttal:

Thank you to the authors for their helpful responses to my questions, and for the additional experiment results (the folding video). Seeing as the proposed method does not use depth and the cameras are not calibrated, I agree with the authors that it is fine to not compare against more data-efficient methods such as EquivAct. I do think the paper is a valuable contribution to the community because it pushes the bounds of how challenging the tasks are which can be solved by these pixel-to-action methods, so I recommend acceptance. However, I still lean more towards Weak Accept than Strong Accept for two reasons.

First, because in their response the authors clarified that the core novel contribution is neither the architecture (present in the ALOHA paper) nor the training algorithm (diffusion policy), but the overall “recipe” for data collection. Part of this is the data collection protocols, which allow even non-expert users to collect demonstrations for solving these tasks. However, if the authors wish to frame this as the main contribution, then this needs an experiment comparing demonstrations collected with and without this protocol (i.e. if the users did not receive any advice from the expert on how to solve the task using teleoperation). As it stands, it is not clear what makes this proposed method any more accessible to non-expert users than the contributions already published in prior work (ALOHA, diffusion policy).

Second, because I am still worried about the future of this line of work. Increasing the number of demonstrations from 6493 to 8658 does not improve performance. In their rebuttal, the authors argue that their method allows for scaling up data collection to a “global labor market”. But the results indicate that this would not be enough to reach satisfactory performance. The results also show that filtering data is also not the answer, as both good and suboptimal demonstrations are required for the best performance. So, the paper would benefit from some ideas about the future of this direction, to show how this line of work can eventually reach production-ready success rates and reduce the amount of human effort required.

**Quality Of The Limitations Section:**

3

**Questions For Rebuttal:**

1. Line 119: would be good to include here how long this training takes as well.
2. Will the authors publish the protocol documents for data collection, as described in Section 3.2? This would be helpful to the research community also working on imitation learning, and for reproducing these results.
3. Who were the 35 non-researcher operators who collected the 26,000 human demonstrations? Are these students, interns, paid contractors? It would be useful to know more about this in order to evaluate how costly it would be for others in the community to reproduce or build on this line of work.

**Robotics Focus:**

4

**Summary Of Paper:**

This paper investigates whether collecting more demonstrations is sufficient to solve challenging dexterous tasks (such as hanging a shirt or tying shoelaces) using end-to-end pixel-to-actions imitation learning. A framework is proposed where a transformer-based policy is trained using a diffusion objective. The challenge of collecting many demonstrations across different robot operators and workstations is addressed. Experiments are conducted including ablation studies, both on difficult real-world tasks and in simulation.

**Summary Of Recommendation:**

I believe that ALOHA Unleashed will be a valuable contribution to the CoRL community, because of the useful scaling results and the impressively challenging tasks addressed. For now I recommend weak accept, conditioned on the authors addressing the issues raised above, such as clarifying the method novelty and comparing against some more data-efficient imitation learning method.

---

### Author Rebuttal · Authors · 2024-08-14

Please view the rebuttal file below and see Official Comments to each reviewer.

---

### Decision · Program_Chairs · 2024-09-04

**Decision:**

Accept

**Comment:**

**Pre-rebuttal**

This paper receives two positive and one strongly positive reviews.

Strengths mentioned:
- Strong relevance (**qSxg**).
- Strong technical quality (**zByN**).
- Impressive demonstrations on long horizon tasks (**zwLY**, **qSxg**).
- Well designed system (**qSxg**).
- Clarity of writing and presentation (**zwLY**).

Weaknesses mentioned:
- Unclear technical contribution (e.g. "reads like a technical report") (**zByN**, **zwLY**, **qSxg**).
- Lack of details on the data collection process (**zwLY**, **qSxg**).
- Lacking in evaluation quality and ablation (**zwLY**).
- More scaling analysis is needed (**zwLY**).
- Concern on further scalability (**qSxg**).

---
**Post-rebuttal**

All the reviewers remain positive after the rebuttal. Overall the paper has delivered interesting findings by challenging the current imitation learning algorithms with more difficult tasks as well as scaling up the data collection effort.

As brought up by **qSxg**, the paper can be benefited by adding some mention about the potential paths for further scaling.

AC also encourages the authors to include the new sim evaluation results from the rebuttal to the paper.

Typos:
- [Line 76]: "Unleashed , " -> "Unleashed, "